# Induction of retinopathy by fibrillar oxalate assemblies

Dor Zaguri[1,8], Shira Shaham-Niv[1,2,8], Efrat Naaman[3,8], Michael Mimouni[3,4], Daniella Magen[4,5], Shirley Pollack[5], Topaz Kreiser[1], Rina Leibu[3], Sigal Rencus-Lazar[1], Lihi Adler-Abramovich[6], Ido Perlman[4], Ehud Gazit[1,7]* & Shiri Zayit-Soudry[3,4]*

The formation of metabolite fibrillar assemblies represents a paradigm shift in the study of human metabolic disorders. Yet, direct clinical relevance has been attributed only to metabolite crystals. A notable example for metabolite crystallization is calcium oxalate crystals observed in various diseases, including primary hyperoxaluria. We unexpectedly observed retinal damage among young hyperoxaluria patients in the absence of crystals. Exploring the possible formation of alternative supramolecular organizations and their biological role, here we show that oxalate can form ordered fibrils with no associated calcium. These fibrils inflict intense retinal cytotoxicity in cultured cells. A rat model injected with oxalate fibrils recaptures patterns of retinal dysfunction observed in patients. Antibodies purified from hyperoxaluria patient sera recognize oxalate fibrils regardless of the presence of calcium. These findings highlight a new molecular basis for oxalate-associated disease, and to our knowledge provide the first direct clinical indication for the pathogenic role of metabolite fibrillar assemblies.

[1] School of Molecular Cell Biology & Biotechnology, Tel Aviv University, Tel Aviv 69978, Israel. [2] Blavatnik Center for Drug Discovery, Metabolite Medicine Division, Tel Aviv University, Tel Aviv 6997801, Israel. [3] Department of Ophthalmology, Rambam Health Care Campus, Haifa 3109601, Israel. [4] Ruth and Bruce Faculty of Medicine, Technion Israel Institute of Technology, Haifa, Israel. [5] Pediatric Nephrology, Rambam Health Care Campus, Haifa 3109601, Israel. [6] Department of Oral Biology, The Goldschleger School of Dental Medicine, Sackler Faculty of Medicine, Tel Aviv University, Tel Aviv 69978, Israel. [7] Blavatnik Center for Drug Discovery for Drug Discovery, Tel Aviv University, Tel Aviv 6997801, Israel. [8] These authors contributed equally: Dor Zaguri, Shira Shaham-Niv, Efrat Naaman *email: ehudga@tauex.tau.ac.il; s_soudry@rambam.health.gov.il

Various inborn error of metabolism (IEM) disorders are characterized by the accumulation of specific metabolites. In many cases, severe developmental abnormalities are associated with the high concentration of metabolites in tissues and organs. Yet, the mechanism leading to the specific symptoms is not fully understood. In various disorders, most notably phenylketonuria, the etiology of the disease was suggested to be associated with the formation of toxic fibrillar supramolecular assemblies by accumulating phenylalanine. Moreover, the deposition of phenylalanine was observed in the brains of phenylketonuria patients post mortem[1]. However, while metabolite crystals are known to be associated with the etiology of various diseases[2,3], direct proof for the causative role of metabolite fibrils in a clinical context has so far been lacking.

Primary hyperoxaluria, a group of IEM diseases, results from abnormal accumulation of oxalate[4]. Specifically, primary hyperoxaluria type 1 typically manifests in childhood with recurrent kidney stones and nephrocalcinosis. Over time, renal failure results in impaired urinary excretion and hence increased serum oxalate levels, leading to the deposition of oxalate aggregates in the eyes, joints, thyroid, and heart[5,6]. The clinical pathological aspects have been attributed to the accumulation of calcium oxalate aggregates, mostly in the form of crystals[3,6–9]. Such crystalline deposits seen in the retina in hyperoxaluria type 1 patients were linked to oxalate retinopathy. Given the rarity of hyperoxaluria, occurring in ~1:120,000 births overall[10], retinal findings reported in the literature are scant and highly variable[11], but include the accumulation of crystalline flecks at the level of the retinal pigment epithelium (RPE) and around retinal blood vessels. Severe visual impairment accompanied by progressive loss of retinal architecture was reported as a possible outcome of the deposition of crystals[12]. However, the underlying mechanism of oxalate retinopathy remains not fully deciphered.

Here, we present the first evidence for a causative role of metabolite fibrillar assemblies in an inborn metabolic disorder. A remarkable discrepancy between a deficient retinal function diagnosed in several pediatric hyperoxaluria type 1 patients and the lack of evident retinal crystals deposition suggested that in addition to crystal-related injury, alternate pathways may play a role in the pathogenesis of oxalate-related retinopathy. We therefore sought to explore the ability of oxalate to self-assemble into alternative ordered structures, and to characterize whether such deposits may harbor biological consequences relevant to the visual impairment observed in the patients. We find that under defined conditions, oxalate forms ordered nanofibrils. Notably, spectroscopic analysis indicates the absence of calcium from the formed fibrils, in marked contrast to the organization of canonical oxalate crystals, in which the metabolite is coordinated with calcium[10]. Moreover, the self-assembled fibrils induce cytotoxicity in cultured cells and cause in vivo retinal impairment when injected into rat eyes, resulting in characteristics similar to those identified in our hyperoxaluria patients. Finally, immunochemistry allows us to specifically identify the oxalate fibrils in sera of several hyperoxaluria patients, but not in healthy control individuals, confirming the occurrence of these specific assemblies during the course of the disease, as well as their immunogenicity. Taken together, our results demonstrate, for the first time to our knowledge, the formation of toxic oxalate fibrillar assemblies and the potential role of these structures in the pathophysiology of retinal oxalosis. These results not only shed new light on the pathology induced by the accumulation of oxalate in hyperoxaluria, but also provide experimental evidence on the pathological role of metabolite fibrillar assemblies in an IEM disorder, with potential implications for human metabolic diseases in general.

## Results

**Impaired retinal function in the absence of crystals deposition.** As part of multi-disciplinary tertiary medical care, five children from three different families with primary hyperoxaluria type 1 underwent comprehensive evaluation of their ophthalmologic status, including retinal clinical imaging and electrophysiological testing. All patients demonstrated entirely normal ophthalmoscopic findings with no apparent retinal structural changes. Specifically, no crystal deposits were identified. Retinal imaging, including spectral domain optical coherence tomography, revealed intact morphology with no evidence of abnormal deposition (Supplementary Fig. 1). Visual evoked potentials (VEP), recorded in response to pattern reversal visual stimuli in all children, showed P100 waves of normal amplitude and latency, indicating normal macular function. The flash VEP responses consisted of waves of normal amplitudes appearing at normal implicit time, indicating normal conductance in the visual pathways from each eye to the primary visual cortex. To quantitatively assess their retinal function, all children underwent full-field electroretinography (ERG) under dark-adapted conditions. Surprisingly, the responses recorded in two children demonstrated a significant reduction of the ERG a-wave and b-wave amplitudes (Fig. 1a), indicating unexpected, impaired global retinal function, in spite of the absence of any retinal crystals or associated morphological abnormalities.

**Self-assembly of oxalate into nanofibrils.** To provide a possible mechanistic explanation for the observed inconsistency between the lack of crystal deposition and impaired retinal function, we aimed to examine the possible in vitro self-assembly of oxalate into alternative supramolecular architectures, other than crystals, as recently shown for several other metabolites[1,13]. For this purpose, a well-established self-assembly methodology was applied, as we have previously described[1,13,14]. Briefly, a supersaturated solution, as previously used to prepare calcium oxalate crystals[15–18], containing ~70 mM calcium oxalate in PBS to reflect physiological pH and ionic strength was heated to 90 °C to increase solubility and ascertain homogenous monomeric solution (Fig. 2a). A differential series of centrifugation and fraction separation steps resulted in two distinct populations of assemblies predominantly consisting of fibrils or crystals (Fig. 2b, c). To obtain predominantly crystals, the supersaturated solution which was heated for several hours, was gradually cooled overnight at room temperature and then centrifuged for 5 min at $20,000 \times g$. Transmission electron microscopy (TEM) of a sample taken from the pellet-supernatant interface showed predominantly crystals (Fig. 2c). To obtain predominantly fibrils, the solution was heated for several hours and centrifuged immediately after heating for 5 min at $20,000 \times g$. The supernatant, containing oxalate monomers, was then collected and allowed to gradually cool overnight. TEM analysis of these samples confirmed the predominance of typical elongated nanofibrillar structures (Figs. 1b and 2b). Energy-dispersive X-ray spectroscopy (EDS) analysis was further performed to examine the presence of calcium in both fractions. Surprisingly, while calcium was clearly identified in the crystals (Fig. 2c), no calcium was detected in the fibrils (Fig. 2b). To further show the fundamental ability of oxalate to self-assemble into fibrillar entities without the presence of calcium, we tested the formation of such structures by sodium oxalate. Indeed, self-assembled oxalate fibrils were obtained, as verified by TEM (Supplementary Fig. 2). The lack of calcium in the oxalate fibrils, as compared to calcium oxalate crystals[19,20], indicates an alternative arrangement of the oxalate building blocks in the newly identified morphology.

**Cellular internalization and cytotoxicity of oxalate fibrils**. To study the possible cytotoxicity of the oxalate fibrillar assemblies in cultured cells, the MTT cell proliferation assay was employed on ARPE-19, an immortalized human cell line with differentiated structural and functional properties characteristic of RPE cells in vivo (Fig. 1c)[21,22]. First, to verify the presence of the oxalate fibrils in cultured ARPE-19 cells treated with oxalate assemblies, antibodies specific for the oxalate fibrils were employed. These antibodies were raised in rabbits, and their specificity towards the fibrillar oxalate organization was validated in vitro (Supplementary Fig. 3a). ARPE-19 cells incubated overnight with oxalate fibrils demonstrated positive staining with the specific antibody, while no such staining could be observed in non-treated cells (Supplementary Fig. 4a, b). Z-stack analysis confirmed the cellular localization of the fibrils (Supplementary Fig. 4c). Next, ARPE-19 cells were cultured with cell growth media containing either oxalate fibrils, calcium oxalate crystals or sodium oxalate fibrils at equivalent concentrations of ~6 mM. Oxalate fibrils, derived either from calcium oxalate solution or from sodium oxalate solutions, conferred the highest toxic effect, reducing cell viability to 60% and 55% respectively, compared to control cells (Supplementary Fig. 4d). Cells cultured with calcium oxalate crystals showed a similar decrease in viability to ~60%. Consistently, unassembled oxalate and alanine, an average-sized amino acid that does not form fibrillar structures even at higher

concentrations (50 mM)[13,23], did not demonstrate a toxic effect (90% and 95% viability, respectively) (Supplementary Fig. 4d). Thus, these results support the notion that the cytotoxicity did not result merely from oxalate concentration, but rather from its supramolecular nanofibrillar structure. To further assess the cytotoxicity of oxalate fibrils on additional cell line representing a target of systemic oxalosis, human embryonic kidney cells (HEK-293) were treated in a similar manner. Oxalate fibrils, derived either from calcium oxalate or from sodium oxalate solutions, conferred the highest toxic effect, reducing cell viability to 55% and 60% respectively, compared to control cells (Supplementary Fig. 4e). Cells cultured with calcium oxalate crystals showed a subtler decrease in viability to ~75%. Unassembled oxalate and alanine did not demonstrate a toxic effect (95% and 90% viability, respectively) (Supplementary Fig. 4e).

**Immunodetection of oxalate fibrils in hyperoxaluria patients**. We next aimed to determine the clinical and pathological relevance of the oxalate fibrillar assemblies by examining their presence in the hyperoxaluria patients. First, we examined the presence of systemic antibodies that can specifically recognize pre-formed oxalate fibrils in serum samples obtained from the five patients manifesting different levels of disease severity (Fig. 2d, e). Antibodies purified from the sera were used as

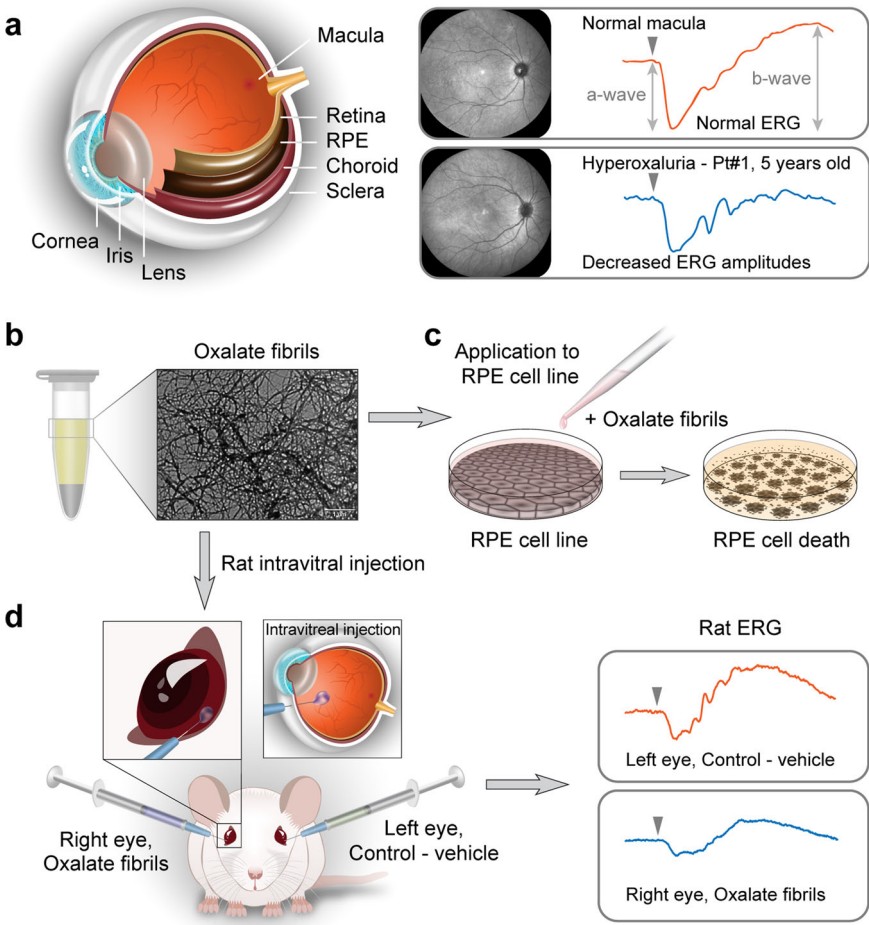

**Fig. 1 Schematic illustration of the experimental system. a** Abnormal electroretinography (ERG) responses were observed in patients in the absence of crystal deposition retinopathy. **b** Formation of an alternative organization of oxalate supramolecular structures in vitro. Fibrillary nano-assemblies were formed along the interface of separated oxalate supersaturated solution, as determined by transmission electron microscopy (TEM). **c** The non-canonical structures inflicted cytotoxicity in retinal pigment epithelium (RPE) cell line. **d** Injection of the oxalate fibrils resulted in diminished ERG responses in treated rat eyes, but not in the corresponding control eyes.

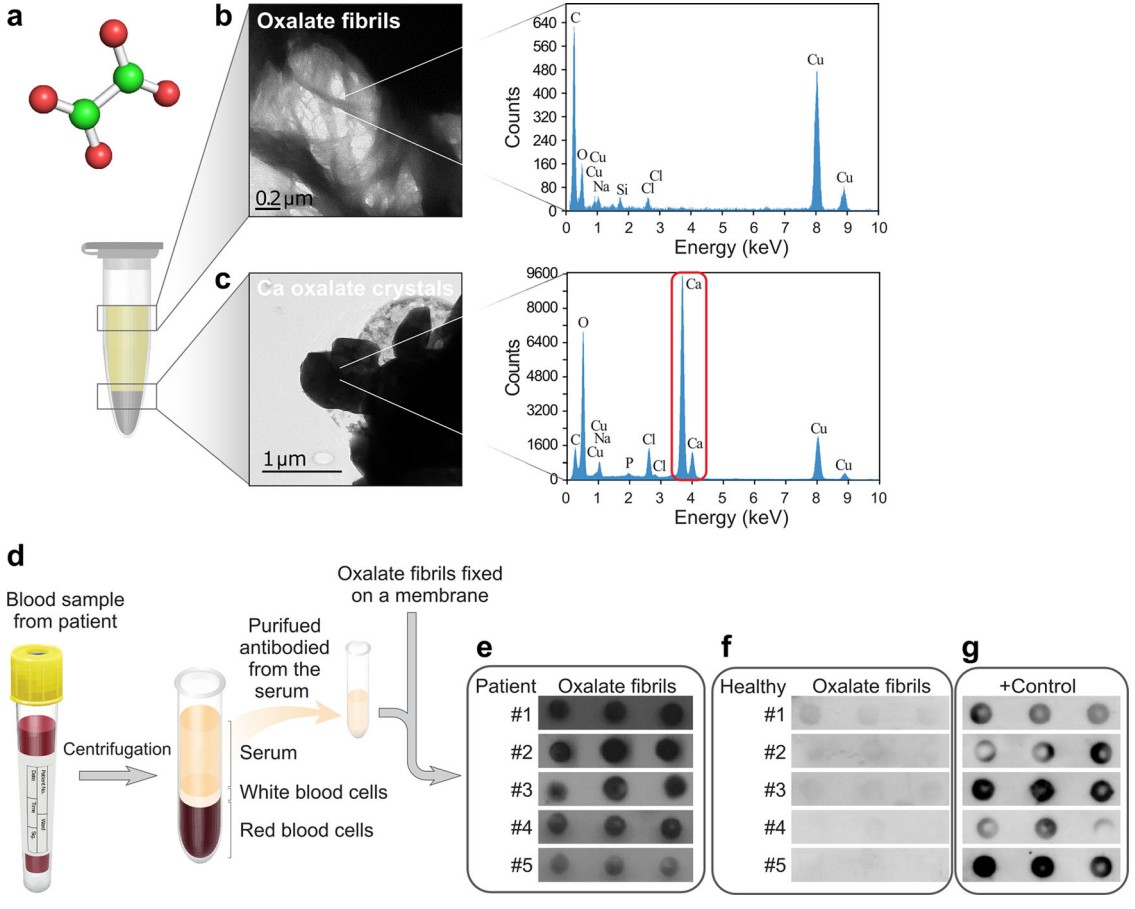

**Fig. 2 Oxalate self-assembly into nanofibrils recognized by antibodies purified from hyperoxaluria patients' sera. a** Chemical formula of oxalate, carbon in green, oxygen in red. **b**, **c** Supersaturated calcium oxalate solution in PBS was heated to 90 °C prior to a differential series of centrifugation and fraction separation steps. This resulted in two distinct populations of assemblies (see "methods" section) visualized by TEM and verified by EDS analysis (the gray lines indicate the selected area examined). The red circle in the calcium oxalate crystals spectra marks the calcium peak. **b** Oxalate fibrils (scale bar: 0.2 μm).
**c** Calcium oxalate crystals (scale bar: 1 μm). **d**–**g** Antibodies were purified from hyperoxaluria patients and healthy control sera. **e** Dot blot assay using preformed oxalate nanofibrils as antigens and antibodies purified from sera obtained from five patients as the primary antibodies. **f** Dot blot assay using preformed oxalate nanofibrils as antigens and antibodies purified from sera obtained from five healthy controls as the primary antibodies. **g** A positive control for antibody titer and purification, demonstrating the response of the antibodies purified from the control sera towards the ubiquitous tetanus antigen.

primary antibodies in a dot blot assay using in vitro assembled oxalate fibrils and unassembled oxalate as antigens. The results demonstrated the detection of preformed oxalate fibrils by purified antibodies from each of the hyperoxaluria patients tested (Fig. 2e), in distinction from unassembled oxalate, which produced a negative signal (Supplementary Fig. 3b). In contrast, antibodies purified from five healthy subjects, serving as negative controls, showed no recognition of oxalate fibrils (Fig. 2f). To verify a proper level of antibody titer in the control samples, detection of the ubiquitous tetanus toxin[24,25] was demonstrated (Fig. 2g). Next, to obtain direct evidence for the systemic presence of oxalate fibrils in hyperoxaluria patients, we searched for oxalate fibrils in the sera using our customized anti-oxalate fibrils antibody (Supplementary Fig. 3). The sera of hyperoxaluria patients were loaded onto a membrane as antigens, and their reaction with the anti-oxalate fibrils antibody was tested in a dot blot assay. The positive recognition in each of the cases tested indicates the presence of oxalate fibrils in the serum of hyperoxaluria patients (Supplementary Fig. 3c). Sera from healthy subjects reacted with the antibody in the same manner produced a negative signal (Supplementary Fig. 3d).

**Oxalate fibrils exert impaired retinal function in rats eyes**. To gain insight into the pathogenicity of oxalate fibrillar assemblies,

we examined their impact on retinal function in vivo. Wild-type rats ($n = 10$) were treated with oxalate fibrils administered via intravitreal injection into one eye, while the fellow eye was treated with the vehicle only and served as control (Fig. 1d). Notably, while the retinal morphology of all the experimental eyes appeared ubiquitously normal upon inspection, all rats exhibited diminished ERG amplitudes in the treated eye compared to the control eye as soon as 7 days after the injection (Fig. 3a). To quantify the impact of the fibrils on the retinal function in each rat, the ratio between the ERG a-wave and b-wave maximal amplitudes (Vmax) from the experimental versus control eye was calculated for each time point (Fig. 3b)[26–28]. Similarly, the ratio between the Vmax values of the experimental and control eyes was calculated for the entire experimental group at each recording session (Fig. 3c). The derived Vmax ratios were approximately 0.65 for both the a-wave and the b-wave throughout the follow-up period ($p < 0.01$ for all), indicating a significant decline in retinal function in the experimental eyes compared to the control eyes. The oxalate fibrils-mediated reduction in retinal function was apparent already 7 days after injection and persisted for at least 30 days. To further validate the basis of the observed retinal toxicity, the impact of non-aggregated oxalate on retinal function was tested. Rats ($n = 3$) treated with monomeric oxalate exhibited intact ERG responses for up to 30 days after the injection

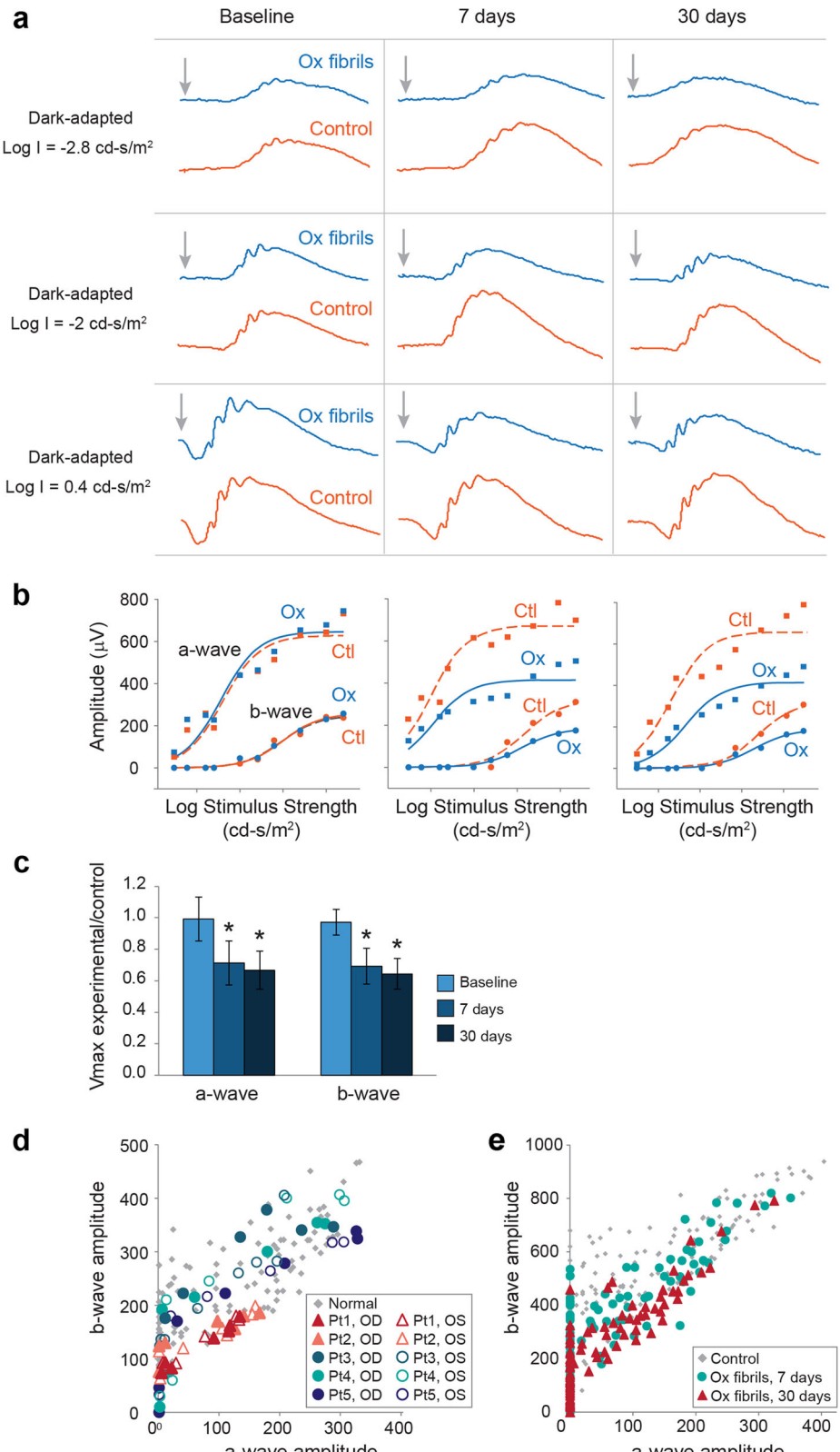

(Supplementary Fig. 5). These results support the notion that the cytotoxic profile of oxalate is derived from its supramolecular fibrillar architecture.

To quantify the deviation in the ERG pattern, as compared to the hyperoxaluria patients, the relation between the amplitude of the a-wave and the b-wave was determined for each response[29]. Figure 3d illustrates the normal range of measures from 15 healthy human subjects, versus the corresponding recordings from the examined patients. The ERG responses from patients 1 and 2 were characterized by subnormal amplitudes, with the plot of the b-wave versus the a-wave showing a marginally low value. These measures indicated that the subnormal ERG amplitude resulted mainly from reduced photoreceptor response, accompanied, to a lesser extent, by impairment of signal transmission

**Fig. 3 The electroretinographic responses in rats treated with oxalate fibrils are similar to those observed in hyperoxaluria patients. a** Dark-adapted electroretinographic (ERG) responses elicited by flashes of different strengths, as denoted in log units to the left, are shown for each recording session. Arrow: time of light stimulus. Calibration bar: 200 µV. **b** Response-stimulus strength relationships for the dark-adapted ERG responses of the same rat. The curves were fitted to a hyperbolic type function. **c** The ratio between experimental and control Vmax of the ERG a-wave and b-wave at three time points, as indicated. Error bars: standard deviation of the mean. **d**, **e** The relationship between the ERG b-wave and a-wave amplitude was plotted for each response evoked in the dark-adapted state in hyperoxaluria patients and in model rats. In each frame, the light gray markers denote the relationship obtained in normal eyes. **d** Data obtained from 5 hyperoxaluria patients and 15 healthy controls. **e** Data from the experimental rat eyes treated with intravitreal injection of oxalate fibrils are illustrated for each recording session after the injection.

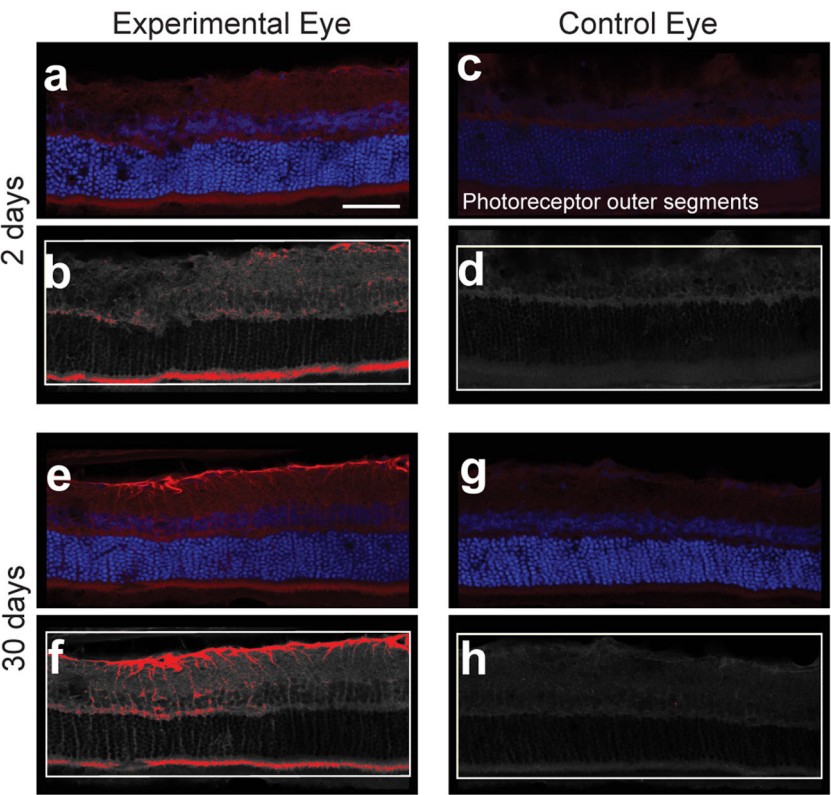

**Fig. 4 Oxalate fibrils are present in outer retinal layers and in retinal glial cells in treated eyes.** Retinas of rats injected with oxalate fibrils were stained with a specific antibody and examined using confocal microscopy. For each section, the microscopy image is shown together with the corresponding analysis of fluorescence intensity. **a**, **b** Retinal section from an experimental eye 2 days after the injection demonstrates scattered fluorescence along the inner retinal layers, with an intense band-shaped signal along the photoreceptors outer segments. **c**, **d** The corresponding retinal section from the control eye. **e**, **f** 30 days after the injection, the experimental retina demonstrates intense fluorescence in vertical figures with a typical morphology for Müller glial cells. **g**, **h** Control eye of the same rat. Scale bar: 25 µm.

within the retina. Remarkably, the marginally electronegative pattern of the ERG responses seen in the experimental rats, as characterized by the decreased b-wave to a-wave amplitude ratio at 30 days after the injection (Fig. 3e), was strikingly similar to the abnormal configuration observed by clinical electrophysiology testing of the patients (Fig. 3d).

**Immunostaining of oxalate fibrils in treated rats retina sections**. To characterize the presence of oxalate nanofibrils in the tissue specimens, retina sections from the experimental and controls eyes of the treated rats were subjected to immunohistochemistry analysis using the oxalate-fibrils specific antibody (Fig. 4). In agreement with the ERG findings, indicating compromised photoreceptor function, intense staining was detected along the photoreceptors outer segments as soon as 2 days post injection (Fig. 4a-d), and persisted 30 days thereafter (Fig. 4e, f). Interestingly, 30 days after the injection, an intense staining of Müller macroglial cells by the oxalate fibrils antibody was detected in the experimental but not in the control retinas

(Fig. 4e-h). Immunoreactivity for glial fibrillary acidic protein (GFAP), a marker of retinal Müller glial cell activation, demonstrated positive staining of Müller cells in the experimental retinas, but not in the control eyes (data not shown). In contrast, neither oxalate fibrils nor GFAP were detected in the control eyes (Fig. 4 and data not shown).

**Discussion**
Our study presents the first evidence, to our knowledge, for the role of metabolite fibrillar structures as a causative agent in a human metabolic disorder. Driven by puzzling clinical findings in pediatric hyperoxaluria patients, we set out to comprehensively explore an alternative underlying biomolecular mechanism for the associated retinopathy, other than the deposition of calcium oxalate crystals[10]. First, we demonstrated the in vitro molecular self-assembly of oxalate into fibrillar structures at the nano-scale. These ordered assemblies, shown by EDS to contain no calcium, represent a distinct immunological entity, as confirmed by their identification by antibodies purified from patient sera. The

ordered oxalate fibrils exerted intense cytotoxicity, which was evident in cultured RPE cells, as well as in rat eyes treated with intravitreal administration. Notably, the retinal dysfunction inflicted in rats was substantially similar to the abnormal retinal function noted in clinical ERG of hyperoxaluria patients. The retinotoxic impact of the fibrils was further evident by their internalization into retinal Müller cells in treated rat eyes. Based on the resemblance of the oxalate fibrils, as reflected by their fibrillar morphology, cytotoxicity and triggering of the immune system, to other metabolite fibrils reported in our previously published articles on metabolite amyloid assemblies[1,13,14,23,30,31], we hypothesize that the oxalate fibrils might also comprise similar ultrastructural properties. We hope that the observations presented here will promote further exploration of the supramolecular structure of the oxalate assemblies.

Our in vivo rat model reliably recapitulates the clinical features observed in the patients, establishing the causative role of oxalate fibrils in hyperoxaluria-associated retinopathy. In accordance, we and others have previously shown that at pathological concentrations, the phenylalanine single amino acid self-assembles into well-ordered fibrils showing cytotoxic properties that likely play a role in phenylketonuria[1,32]. Later, we demonstrated that several additional metabolites accumulating in inherited systemic disorders can form ordered non-proteinaceous fibrils, which induce apoptotic cell death in cultured cells[13]. These studies led us to postulate a unified new dogma of metabolite self-assembly into ordered structures, suggesting their key role in human disease mechanisms[33].

Our current study provides the first in vivo evidence for this new paradigm, depicting a pivotal role of ordered oxalate fibrils in the initiation of the retinal phenotype clinically observed in hyperoxaluria patients. The remarkable similarity in the abnormal retinal parameters between human patients and injected rats provides a compelling evidence for the suggested pathological role of these assemblies. This newly established in vivo model lays the basis for the development of future therapeutics targeting the assembly of oxalate into cytotoxic fibrillar assemblies. Furthermore, the causative role of self-assembled metabolite fibrillar structures demonstrated here can be further explored as a common underlying biomolecular mechanism in diverse inborn error of metabolism disorders.

## Methods

**Patients and clinical evaluation.** Included in the study were children aged 0–18 years, previously diagnosed with primary hyperoxaluria type 1 on the basis of clinical criteria and confirmatory genetic testing, who are treated at the Institute of Pediatric Nephrology of the Ruth Rappaport Children's Hospital, at Rambam Health Care Campus, Haifa, Israel. The study protocol was reviewed and approved by the Institutional Review Board of the Rambam Health Care Campus, Haifa, Israel. All data for this study was collected and analyzed in accordance with the policies and procedures of the Institutional Review Board of the Rambam Health Care Campus and the tenets of the Declaration of Helsinki. Written informed consent was obtained from the participants or their parents.

All children underwent a complete, as possible, ophthalmic assessment, including best-corrected visual acuity and slit-lamp examination of the anterior segment. The parents were thoroughly questioned to assess for visual disturbances. Pharmacological pupil dilation with topical tropicamide facilitated an ophthalmoscopic examination and ophthalmic imaging, including fundus autofluorescence and macular spectral domain-optical coherence tomography (SD-OCT) (Spectralis HRA + OCT Laser Scanning Camera, Heidelberg Engineering, Vista, CA). To evaluate the functional status of the retina, full-field electroretinography (ff-ERG) recordings were performed after 20 min of dark adaptation using the UTAS 3000 electrophysiological system (LKC Technologies, Gaithersburg, MD), in accordance with the International Society for Clinical Electrophysiology of Vision (ISCEV) standards[26]. The responses were simultaneously recorded from both eyes using DTL Plus electrodes (diagnosys LLC). Electroretinogram analysis was based on amplitude and implicit time measurements. The a-wave amplitude was measured from the baseline to the trough of the negative wave. The b-wave amplitude was measured from the trough of the a-wave to the peak of the b-wave.

Visual evoked potential (VEP) testing was carried out using 2 electroencephalographic electrodes. The active electrode was placed approximately 1 cm above the inion, and the reference electrode was placed on the forehead. An ear clip served as the ground electrode. The signals were amplified and filtered (1–100 Hz) (UTAS-4000, LKC Technologies, Gaithersburg, USA). Fifty responses were averaged for each recording. Pattern reversal VEP was performed as long as the child could cooperate and retain a stable fixation. At a viewing distance of 1 m, stimulus check sizes of 9–144 arcminutes were presented for each eye separately as well as for both eyes simultaneously. For VEP analysis, the shape of the waveform, the timing of the peak and trough, and the amplitude of the negative and positive waves were measured.

For detection of systemic antibodies against oxalate fibrils, a single blood sample (0.5–1 ml) was drawn by venipuncture of a peripheral vein.

**Preparation and biophysical characterization of oxalate fibrils and crystals.** Oxalate structures were obtained using heating and cooling, a well-established self-assembly method, as we have previously described[1,13,14]. The specific protocols used to obtain each structural form are outlined below.

Crystals: Fresh 10 mg/ml stock solution of calcium-oxalate monohydrate diluted in phosphate buffered saline (PBS) was heated for 4 h at 90 °C to obtain a supersaturated solution, before gradual cooling overnight at room temperature. The following day, the solutions were centrifuged for 5 min at 20,000 × g and separated to three distinct phases: supernatant (sup), pellet and sup/pellet interface. A crystals-containing fraction was collected from the interphase and was transferred into a new tube (Fig. 2c).

Fibrils: Fresh 10 mg/ml stock solution of calcium-oxalate monohydrate diluted in PBS was heated for 4 h at 90 °C to obtain a supersaturated solution and then immediately centrifuged for 5 min at 20,000 × g for separation into three distinct phases: supernatant (sup), pellet and sup/pellet interface. The sup fraction, containing only the soluble phase, was collected and transferred into a new tube, followed by overnight gradual cooling at room temperature to form fibrils (Fig. 2b).

Unassembled oxalate- 2 mg/ml oxalate dissolved in PBS. The soluble phase was used a unassembled oxalate.

Sodium oxalate fibrils: 2 mg/ml solution of sodium oxalate diluted in PBS was heated for 4 h at 90 °C to obtain monomeric state, before gradual cooling overnight at room temperature to form fibrillar structures.

**Transmission electron microscopy.** A 10 μl aliquot was placed on 400-mesh copper grids. After 2 min, excess fluids were removed. Samples were viewed using a JEM-1400Plus electron microscope operating at 80 kV.

**Generation of anti-oxalate fibrils antibodies.** Oxalate fibrillar assemblies, obtained as outlined above, served as antigens in a series of immunization cycles in rabbits (Hy-labs, Rehovot, Israel). Polyclonal IgG antibodies were purified using a protein G column chromatography. Dot blot binding assay was employed to validate the specificity of the antibody to the fibrils.

**Antibody purification from human sera.** Human serum samples were diluted 1:5 in PBS before loading onto a 2 ml protein G column (protein G beads – A2S, gravity column, Thermo scientific), with the flow-through collected and reloaded 3 times. Bound antibody was eluted with 0.1 M glycine buffer (pH 3.0). The elution was neutralized by adding 20% v/v 1 M Tris–HCl (pH 7.3). Next, protein-containing fractions were pooled and buffer exchange into PBS was performed by centrifugal filtration with 50KD cutoff (Amicon, Merck). Antibodies concentration was determined using the BCA reagent (Sigma-Aldrich). Antibody purity was assessed by SDS-PAGE.

**Dot-blot immunoassay.** Freshly prepared stock solution of pre-formed oxalate nanofibrillar assemblies was used as the antigen. Bio-Dot microfiltration apparatus was used for fixation on a Nitrocellulose (NC) membrane (GE–healthcare). Oxalate assemblies (200 μl) were loaded in triplicates onto a NC membrane and absorbed overnight at 4 °C. The membrane was blocked with 5% skim-milk. Primary antibody was diluted 1:200. Secondary antibody was diluted 1:10,000. Blocking and primary antibody steps were performed for 2 h each, while the secondary antibody was incubated for 1 h. Between the stages, three cycles of 5 min washes with TBS-T 0.1% were performed. Blots were developed using Luminata Forte Western HRP substrate and visualized using amersham imager 600 (GE).

**Internalization of oxalate fibrils by ARPE-19 cells.** Cultured ARPE-19 cells were grown to 50% confluence on poly-L-lysine coated cover-slips (Mercury) in DMEM/Nutrient Mixture F12 (Ham's) (1:1) medium containing 10% fetal bovine serum (FBS) (Biological Industries, Israel) in 24-well plates. The cells were subsequently treated either with medium containing preformed oxalate fibrillar assemblies or with medium without oxalate that was processed in the same manner. The cells were then rinsed with PBS and fixed in 4% PFA (EMS) for 15 min at room temperature, before being washed twice with 4 °C PBS and treated with 0.25%

Triton X-100 for 10 min at room temperature to allow cellular permeabilization. After thoroughly washing the cells, blocking was performed using 1% Bovine serum albumin (BSA) (Amresco) overnight at 4 °C. Then, the cells were stained using the rabbit polyclonal anti-oxalate fibrils antibody diluted 1:100 in blocking solution for 1 h at room temperature. The cells were washed three times with PBS and anti-rabbit Cy3-conjugated secondary antibody (Jackson immuno-research) diluted 1:200 in blocking solution was added for another 30 min at room temperature in the dark. Finally, cells were washed three times and the cover slips were mounted using 15 μL Vectashield Antifade Mounting Medium with DAPI (Vector laboratories). Imaging was performed using SP8 inverted confocal microscope (Leica Microsystems, Wetzlar, Germany). Excitation/emission wavelengths were 412/450 nm for DAPI and 548/561 nm for Cy3.

**Cell viability assay**. ARPE-19 cells and HEK-293 cells ($2 \times 10^5$ cells/ml) were cultured in 96-well tissue microplates (100 μl per well) and allowed to adhere overnight at 37 °C. Oxalate assemblies were formed in vitro as outlined above, except the solvent was DMEM/Nutrient Mixture F12 (Ham's) (1:1) (Biological Industries, Israel) without FBS. The concentrations of all oxalate preparations were approximately 6 mM. Only half of each plate was seeded with cells. Medium with no oxalate, which was treated in the same manner, was used as a negative control. Medium (100 μl) with or without oxalate assemblies was added to each well. After overnight incubation at 37 °C, cell viability was evaluated using 3-(4, 5-dimethyl-thiazolyl-2)-2,5-diphenyltetrazolium bromide MTT cell proliferation assay (reagent purchased from Sigma, Rehovot, Israel), according to the manufacturer's instructions. Briefly, 10 μl of 5 mg/ml MTT reagent dissolved in PBS was added to each of the 96 wells, followed by a 4 h incubation at 37 °C. Then, 100 μl of extraction buffer (20% SDS dissolved in a solution of 50% dimethylformamide and 50% DDW) was added to each well, followed by 30 min incubation at 37 °C in the dark. Finally, color intensity was measured using an ELISA plate reader at 570 nm and background subtraction at 650 nm. The results represent three biological repeats; the data are presented as mean ± SD.

**Animals**. All animals were treated in accordance with the ARVO statement for the Use of Animals in Ophthalmic and Vision Research and according to institutional guidelines. The Ethics Committee of the Ruth Rappaport Faculty of Medicine, Technion, approved the study protocol. The rats were housed under 12/12-hour light/dark cycles, with unrestricted access to food and water. Prior to the intravitreal injection and electrophysiological tests, adult male Sprague-Dolly (SD) rats were anesthetized with intramuscular injection (0.5 ml/kg) of a mixture consisting of Ketamine hydrochloride (10 mg/ml), Acepromazine maleate 10% and Xylazine 2% at a 10:2:3 proportion, respectively. Cyclopentolate hydrochloride 1% was used for full pupil dilation and topical Benoxinate HCl 0.4% was instilled for topical anesthesia. The rats were then intravitreally injected with 10 μl of preformed oxalate fibrils in PBS to the right (experimental) eye, whereas the left (control) eye was treated with the same volume of a blank solution containing PBS. Briefly, a well-trained physician performed the injections using a 32-gauge needle inserted 1 mm posterior to the limbus, while employing indirect ophthalmoscopy to assure proper localization of the needle in the vitreous cavity. Indirect ophthalmoscopy was repeated following the injection to rule-out unintentional damage to the retina or the lens. ERG was performed at baseline, as well as 7 and 30 days after injection. Thirty days following treatment, the rats were sacrificed and the retinas were prepared for immunohistochemistry.

**ERG**. For ERG analysis, the amplitude of the a-wave and the b-wave in the experimental and control eyes of each rat were measured, and their maximal amplitudes ($V_{max}$) were calculated using a hyperbolic type function[27]. The ratios between the maximal amplitudes of the ERG a-wave and b-wave obtained from the experimental and control eye in each rat were calculated. Technical factors potentially affecting the assessment of retinal function, such as the depth of the anesthesia or the duration of the dark adaptation, were thereby minimized[28]. To quantitatively assess the change in the ERG, the relationship between the amplitude of the a-wave and the b-wave was determined for each response from each eye. The dependence of the b-wave on the a-wave was used as an index for signal transmission in the retina[29].

**Immunohistochemistry of rat retinal sections**. The rats were killed up to 5 weeks after injection. The eye was extracted and soaked for up to 1 h in a solution of 2% paraformaldehyde and 2.5% glutaraldehyde in 0.1 M phosphate buffer (pH 7.4). For immunostaining, the eyecup was washed in 0.1 M PBS before being cryoprotected overnight in increasing concentrations of sucrose at 4 °C. The tissue was embedded in OCT and cut into 16-μm thick sections (Reichard Jung microtome). The cryostat sections were blocked in normal non-immune serum (3% serum + 0.1% TritonX-100 + PBS 0.1 M), followed by overnight incubation in a moist chamber at 4 °C with a rabbit anti-oxalate fibrils antibody (raised in this study) diluted 1:500. The secondary antibody was Rhodamine Red-X labeled anti rabbit (donkey anti rabbit, 1:500, Jackson Immunoresearch, West Grove, PA, USA). DAPI was added at 1:2000. Samples were examined using a Zeiss (Oberkochen, Germany) confocal LSM 700 microscope. For immunostaining analysis, the images

were analyzed using Fiji software. Retinal areas were marked, and the fluorescence intensity was measured using the threshold tool.

## Data availability

The data that support the findings of this study are available from the authors on reasonable request; see author contributions for specific data sets.

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

## Acknowledgements
This research work was supported by the Israel Science Foundation (grant no. 802/15; E.G.), The Adelis Forever Foundation (E.G.), and by the Rambam-Ofakim Research Program (S.Z.S.). (D.Z.) would like to acknowledge Dafna Greitzer, Zohar Arnon and Idan Ditchi for graphical editing and assistance in the figuers preparation process. (D.Z.) would also like to acknowledge the Nehemia Levtzion for Excellent Ph.D. Student from Periphery Regions Scholarship.

## Author contributions
D.Z., S.S.N., E.N., M.M. and T.K., all took part in desining and conducting the in-vitro and in-vivo experiments, analyzing the data and writing the manuscript. D.M., S.P., R.L. and I.P. all took part in obtaining and analyzing the clinical data, L.A.A. and S.R.L. conducted the background research and assisted with writing the manuscript. E.G. and S.Z.S., both initiated the study with the conceptual design and planning of the research hypothesis. They designed the experimental system, took part in experimental and clinical data analysis, preparation of figures and manuscript preparation

## Competing interests
All authors declare no competing interests.
