## [Peer Review File · Communications Chemistry]

Reviewers' comments:

Reviewer #1 (Remarks to the Author):

Comment to the authors

The manuscript supports a concept of metabolite fibrillar structures as a consequence of increased single metabolite concentrations in IEM. The authors have shown several lines of evidence that oxalate can form self assembled fibrillar structures and that these fibrils may contribute to the retinopathy in primary hyperoxaluria. All in all, the manuscript is well written and the experimental part of this elaborate and well-controlled study sounds solid.

In the past years, the authors could already show that high concentrations of single metabolites, e. g. phenylalanine, can lead to self assembled fibrillar structures. However, there was no report, at least to the reviewer's knowledge, that showed clear evidence fibrils mechanistically contribute to the phenotype of IEM. The novelty of this study lies in the identification of metabolite fibril-specific antibodies in human specimens and in the fact that artificially generated fibrils, which were administered to rat eyes, could mimic a specific phenotype of retinopathy of primary hyperoxaluria.

In summary, the I do not see any weaknesses of the manuscript and I support a publication without any further review.

Reviewer #2 (Remarks to the Author):

The manuscript by Dr. Zayit-Soudry and colleagues describes a novel pathological mechanism associated with oxalate accumulation in Primary Hyperoxaluria Type I (PH1), a rare genetic condition due to the lack of efficient glyoxylate detoxification in the liver. The main symptoms of PH1 are related to increased urinary excretion of oxalate, a human metabolic end-product. The authors combine in vivo and in vitro data demonstrating that oxalate accumulation leads to the formation of supramolecular assemblies in the form of fibrils, which are responsible for a cellular damage at retinal level. The results obtained are very interesting and innovative in the field of hyperoxaluria, and they give a new view of the mechanisms underlying the pathogenesis of inherited enzymatic deficits. However, the paper requires some major revisions before being considered for publication. In particular, some of the conclusions need additional evidence and controls.

- In the Introduction section, the authors should acknowledge that the deposition of oxalate at retinal level is only observed under conditions of systemic oxalosis, when oxalate accumulates in plasma because of an impaired urinary excretion due to renal failure.

-Did the authors formulate any hypothesis on the ultrastructure of the oxalate fibrils?

-Fig. 2 panel d. No negative control of the reaction of the patients serum with soluble monomeric oxalate or oxalate crystals is provided. Moreover, the presence of antibodies is considered indicative of the presence of the fibrils. If fibrils are formed from oxalate accumulating in plasma, the plasma of patients, and not from healthy individuals, should respond to the antibody against fibrils that the authors produced from immunized rabbits.

-The differences in the red signal between panels a and b of Fig. S4 is not visible

-The authors claim that cytotoxicity is due to oxalate supramolecular assemblies using as control soluble alanine. However, monomeric oxalate represents the negative control to clearly demonstrate that the toxicity is not due to the molecule per se, but rather to its aggregated form.

The same conclusions can be applied to the experiments in rats, where no controls injection using monomeric oxalate has been performed.

-Did the authors test the cytotoxicity of the fibrils against other cellular types? It is known that under systemic oxalosis conditions, oxalate deposits in many tissues, including cardiac muscle.

-The panel S4e is mentioned in the legend but is lacking in the figure

- No panel g is present in figure 2.

Response to the reviewers' comments

We would like to thank the reviewers for their meticulous evaluation, comments and suggestions to improve our manuscript. Following is a list of the changes accordingly made in the revised manuscript. We believe that the revised manuscript indeed provides a better and clearer account of the reported experimental observations.

Response to Reviewer 1:

Remarks to the Author: The manuscript supports a concept of metabolite fibrillar structures as a consequence of increased single metabolite concentrations in IEM. The authors have shown several lines of evidence that oxalate can form self-assembled fibrillar structures and that these fibrils may contribute to the retinopathy in primary hyperoxaluria. All in all, the manuscript is well written and the experimental part of this elaborate and well-controlled study sounds solid. In the past years, the authors could already show that high concentrations of single metabolites, e. g. phenylalanine, can lead to self-assembled fibrillar structures. However, there was no report, at least to the reviewer's knowledge, that showed clear evidence fibrils mechanistically contribute to the phenotype of IEM. The novelty of this study lies in the identification of metabolite fibril-specific antibodies in human specimens and in the fact that artificially generated fibrils, which were administered to rat eyes, could mimic a specific phenotype of retinopathy of primary hyperoxaluria. In summary, I do not see any weaknesses of the manuscript and I support a publication without any further review.

Response: We thank the referee for the positive assessment and for highlighting the novelty in our detailed work. We appreciate the reviewer's comments and are delighted with the stated support in publishing this work in *Communications Chemistry*.

Response to Reviewer 2:

Remarks to the Author: The manuscript by Dr. Zayit-Soudry and colleagues describes a novel pathological mechanism associated with oxalate accumulation in Primary Hyperoxaluria Type I (PH1), a rare genetic condition due to the lack of efficient glyoxylate detoxification in the liver. The main symptoms of PH1 are related to increased urinary excretion of oxalate, a human metabolic end-product. The authors combine in vivo and in vitro data demonstrating that oxalate accumulation leads to the formation of supramolecular assemblies in the form of fibrils, which are responsible for a cellular damage at retinal level. The results obtained are very interesting and innovative in the field of hyperoxaluria, and they give a new view of the mechanisms underlying the pathogenesis of inherited enzymatic deficits. However, the paper requires some major revisions before being considered for publication. In particular, some of the conclusions need additional evidence and controls.

Response: We thank the reviewer for the positive evaluation of our manuscript and for recognizing the innovative nature of our findings. We appreciate the reviewer's comments are thankful for prompting us to further clarify some important experimental points. We have used these suggestions and incorporated them in the text and figures.

We believe that after discussing and completing the additional experiments as described below, we have significantly improved the presentation of our work.

- In the Introduction section, the authors should acknowledge that the deposition of oxalate at retinal level is only observed under conditions of systemic oxalosis, when oxalate accumulates in plasma because of an impaired urinary excretion due to renal failure.

Response: We accept the reviewer's important comment and have changed the text accordingly. This point was clarified in the revised manuscript as follows:

Page 3, line 45-49: Primary hyperoxaluria, a group of IEM diseases, results from abnormal accumulation of oxalate⁴. Specifically, primary hyperoxaluria type 1 typically manifests in childhood with recurrent kidney stones and nephrocalcinosis. **Over time, renal failure results in impaired urinary excretion and hence increased serum oxalate levels, leading to the deposition of oxalate aggregates in the eyes, joints, thyroid, and heart^{5,6}.**

-Did the authors formulate any hypothesis on the ultrastructure of the oxalate fibrils?

Response: We thank the reviewer for allowing us to elaborate on this important topic. Over the past few years, our group has focused on extensive study of the self-assembly process of various small metabolites and their ability to form amyloid supramolecular structure. In particular, the question of the aggregation mechanism and its manipulation, as well as the biological relevance of the metabolite amyloids, were at the heart of our research. Due to the resemblance between the work in our previously published manuscripts [Adler-Abramovich et al. 2012; Shaham-Niv et al. 2015; Shaham-Niv et al. 2018; Shaham-Niv, et al. 2018; Zaguri et al. 2018] and the study presented here on the self-assembly of oxalate (e.g. the fibrillary morphology, cytotoxic effect and triggering of the immune system), we speculate that the oxalate fibrils might possess a similar ultrastructure. Although this issue is out of the scope of this manuscript, we hope that our observations will encourage others to follow up and further examine this intriguing issue. Following the reviewer's comment, we revised the conclusions section of the manuscript as follows:

Page 13, line 251-256: Based on the resemblance of the oxalate fibrils, as reflected by their fibrillar morphology, cytotoxicity and triggering of the immune system, to other metabolite fibrils reported in our previously published articles on metabolite amyloid assemblies^{1,13,14,23,30,31}, we hypothesize that the oxalate fibrils might also comprise similar ultrastructural properties. We hope that the observations presented here will promote further exploration of the supramolecular structure of the oxalate assemblies.

-Fig. 2 panel d. No negative control of the reaction of the patient serum with soluble monomeric oxalate or oxalate crystals is provided. Moreover, the presence of antibodies is considered indicative of the presence of the fibrils. If fibrils are formed from oxalate

accumulating in plasma, the plasma of patients, and not from healthy individuals, should respond to the antibody against fibrils that the authors produced from immunized rabbits.

Response: We thank the referee for raising this important issue, allowing us to clarify these experimental points and add the requested controls. Accordingly, we have examined the reaction of the patients' serum to soluble monomeric oxalate, as suggested by the reviewer. Indeed, in all cases this resulted in a negative signal, highlighting the lack of immune recognition of monomeric oxalate as an antigenic entity (Supplementary figure 3b).

The manuscript was revised as follows:

Page 8, line 176-180: ... in a dot blot assay using in vitro assembled oxalate fibrils **and unassembled oxalate** as antigens. The results demonstrated the detection of preformed oxalate fibrils by purified antibodies from each of the hyperoxaluria patients tested (Figure 2d), **in distinction from unassembled oxalate, which produced a negative signal (Supplementary figure 3b).**

Next, we have explored the reactivity of our antibodies, which were produced from immunized rabbits, towards patients' serum, as per the reviewer's suggestion. Sera from healthy individuals were tested as a negative control. As can be clearly seen in Supplementary figure 3c, each of the patients' sera produced a strong signal, confirming the presence of oxalate assemblies in the patients' serum. In contrast, only a slight background signal was noted when using sera of healthy individuals (Supplementary figure 3d).

The manuscript was revised as follows:

Page 9 line 183-190: **Next, to obtain direct evidence for the systemic presence of oxalate fibrils in hyperoxaluria patients, we searched for oxalate fibrils in the sera using our customized anti-oxalate fibrils antibody (Supplementary figure 3). The sera of hyperoxaluria patients were loaded onto a membrane as antigens, and their reaction with the anti-oxalate fibrils antibody was tested in a dot blot assay. The positive recognition in each of the cases tested indicates the presence of oxalate fibrils in the serum of hyperoxaluria patients (Supplementary figure 3c). Sera from healthy subjects reacted with the antibody in the same manner produced a negative signal (Supplementary figure 3d).**

-The differences in the red signal between panels a and b of Fig. S4 is not visible

Response: We thank the reviewer for bringing this technical issue to our attention. An image editor software that was used unintentionally resulted in a decrease in the resolution of the figures and reduction of the presented fluorescent signal. The images are now incorporated to the final version in their native form, where the difference is clearly observed. Please see revised Supplementary figure 4.

-The authors claim that cytotoxicity is due to oxalate supramolecular assemblies using as control soluble alanine. However, monomeric oxalate represents the negative control to clearly demonstrate that the toxicity is not due to the molecule per se, but rather to its

aggregated form. The same conclusions can be applied to the experiments in rats, where no controls injection using monomeric oxalate has been performed.

Response: We thank the referee for raising this important issue, thereby allowing us to add the requested suitable controls. The cytotoxicity of monomeric oxalate was tested in two cell lines and *in-vivo*. In both tested experimental settings, no effect was exerted by monomeric oxalate. ARPE19 and HEK293 cells were cultured with the different oxalate assemblies and with the unassembled monomeric oxalate. As described before, culturing cells with oxalate assemblies leads to a decrease in cell viability. On the other hand, unassembled monomeric oxalate was found to bear no cytotoxic effect on both cell lines.

Moreover, to further validate the impact of non-aggregated oxalate *in vivo*, rats were treated with intravitreal administration of monomeric oxalate into the right eye, whereas the left eye was injected with a similar volume of the vehicle. In all rats (n=3), no deficit of the ERG responses was noted up to 30 days after the injection. Thus, the results of these additional experiments support the conclusion that the cytotoxic profile of oxalate is determined by its supramolecular architecture.

The manuscript was accordingly revised as follows:

Page 9, line 206 to page 10 line 210: **To further validate the basis of the observed retinal toxicity, the impact of non-aggregated oxalate on retinal function was tested. Rats (n=3) treated with monomeric oxalate exhibited intact ERG responses for up to 30 days after the injection (supplementary figure 5). These results support the notion that the cytotoxic profile of oxalate is derived from its supramolecular fibrillar architecture.**

-Did the authors test the cytotoxicity of the fibrils against other cellular types? It is known that under systemic oxalosis conditions, oxalate deposits in many tissues, including cardiac muscle.

Response: We accept this important comment and thus have examined the ability of oxalate aggregates to cause cytotoxicity in HEK-293, derived from human embryonic kidney cells and commonly used in cytotoxicity experiments. Furthermore, we have added monomeric oxalate to the cell experiments, serving as a negative control. As could be observed in the revised Supplementary figure 4, decreased cell viability was indeed found in HEK-293 cells treated with oxalate fibrillary assemblies, while no cytotoxicity was demonstrated in cells treated with both negative controls, alanine and monomeric oxalate. These observations imply an extensive and more generalized cytotoxic effect induced by oxalate aggregates, supporting its potential pathogenic role in several body systems. The manuscript was revised as follows:

Page 8 line 163-170: **To further assess the cytotoxicity of oxalate fibrils on additional cell line representing a target of systemic oxalosis, human embryonic kidney cells (HEK-293) were treated in a similar manner. Oxalate fibrils, derived either from calcium oxalate or from sodium oxalate solutions, conferred the highest toxic effect, reducing cell viability to 55% and 60% respectively, compared to control cells (Supplementary figure 4e). Cells cultured with calcium oxalate crystals showed a subtler decrease in viability to**

approximately 75%. Unassembled oxalate and alanine did not demonstrate a toxic effect (95 and 90% viability, respectively) (Supplementary figure 4e).

-The panel S4e is mentioned in the legend but is lacking in the figure
- No panel g is present in figure 2.

Response: We are grateful to the reviewer for drawing our attention to these technical issues. We have now appropriately edited Supplementary figure 4 with the legend and relevant text (as detailed above).

REVIEWERS' COMMENTS:

Reviewer #2 (Remarks to the Author):

The authors addressed all the point raised. The manuscript is now acceptable for publication